DoAP2/ERF89 activated the terpene synthase gene DoPAES in Dendrobium officinale and participated in the synthesis of β-patchoulene

Li Decong 1
Liu Lin 1
Li Xiaohong 1
Wei Guo 2
Cai Yongping 1
Sun Xu 1 43449945@qq.com
Fan Honghong 1 hhfan0551@126.com
1 School of Life Sciences, Anhui Agricultural University , Hefei, Anhui , China
2 College of Horticulture and Landscape Architecture, Yangzhou University , Yangzhou, Jiangsu , China
Irfan Mohammad
Electronic publication date: 2024 Jan 18
Publication date: 2024
Volume: 12
Electronic Location ID: e16760
Received 2023 Mar 7; Accepted 2023 Dec 13
Copyright: © 2024 Li et al.
Copyright year: 2024
Copyright holder: Li et al.
License: This is an open access article distributed under the terms of the Creative Commons Attribution License, which permits unrestricted use, distribution, reproduction and adaptation in any medium and for any purpose provided that it is properly attributed. For attribution, the original author(s), title, publication source (PeerJ) and either DOI or URL of the article must be cited.
License URL: https://creativecommons.org/licenses/by/4.0/

Keywords: Terpenoid synthesis, AP2/ERF transcription factor, Dendrobium officinale, Terpene synthase, Transcriptional regulation

Funding: Anhui Natural Science Foundation 1908085MC59 Hefei Natural Science Foundation 2022036 Anhui Provincial Engineering Technology Research Center for Development & Utilization of Regional Characteristic Plants This work was supported by the Anhui Natural Science Foundation (1908085MC59), the Hefei Natural Science Foundation (2022036), and the Anhui Provincial Engineering Technology Research Center for Development & Utilization of Regional Characteristic Plants. The funders had no role in study design, data collection and analysis, decision to publish, or preparation of the manuscript.

==============================
Dendrobium officinale Kimura et Migo is a tonic plant that has both ornamental and medicinal properties. Terpenoids are significant and diverse secondary metabolites in plants, and are one of the important natural active ingredients in D. officinale. The AP2/ERF gene family plays a major role in primary and secondary metabolism. However, the AP2/ERF transcription factor family has not been identified in D. officinale, and it is unclear if it is involved in the regulation of terpenoid biosynthesis. This study identified a sesquiterpene synthetase-β-patchoulene synthase (DoPAES) using transcriptome and terpenic metabolic profile analyses. A total of 111 members of the AP2/ERF family were identified through the whole genome of D. officinale. The tissue-specific expression and gene co-expression pattern of the DoAP2/ERF family members were analyzed. The results showed that the expression of DoPAES was highly correlated with the expression of DoAP2/ERF89 and DoAP2/ERF47. The yeast one-hybrid (Y1H) assays and dual-luciferase experiments demonstrated that DoAP2/ERF89 and DoAP2/ERF47 could regulate the expression of DoPAES. The transcriptional regulatory effects were examined using homologous transient expression of DoAP2/ERF89 in protocorms of D. officinale. DoAP2/ERF89 positively regulated the biosynthesis of β-patchoulene. This study showed that DoAP2/ERF89 can bind to the promoter region of DoPAES to control its expression and further regulate the biosynthesis of β-patchoulene in D. officinale. These results provide new insights on the regulation of terpenoid biosynthesis.

Introduction

Terpenoids are the most significant and diverse secondary metabolites of plants and have pharmacological and biological benefits for humans. More than 80,000 distinct terpenoids exist in plants (Christianson, 2017). Terpenoids are crucial for plant growth and development including fruit ripening, pollinator attraction, and resistance to external challenges (Kessler & Baldwin, 2001). According to Loreto et al. (2014), low-molecular-weight terpenoids that are volatile or semi-volatile, like isoprene, monoterpenoids, sesquiterpenoids, and diterpenoids, may protect plants from abiotic stress both above and below ground (Yonekura-Sakakibara & Saito, 2009). TaPS (β-Patchoulene Synthease) is overexpressed in Arabidopsis thaliana and successfully produces β-patchoulene to verify the biochemical function of TaPS in plants. Transgenic Arabidopsis plants resist herbivores by repelling the larvae of beetroot armyworms, demonstrating the anti-herbivorous activity of β-patchoulene. The catalytic mechanism of TaPS can be explored through homological modeling and site-directed mutagenesis (Pu et al., 2019).

The AP2/ERF transcription factors are a large group of factors that mostly originated in plants. Each member of the family includes an AP2 domain, which is essential to the life cycle of plants (Feng et al., 2020). Numerous plants, such as Hordeum vulgare, Salvia miltiorrhiza, Syntrichia caninervi, and Triticum aestivum, have been identified as having more than 100 members of the AP2/ERF transcription factor family (Guo et al., 2016; Ji et al., 2016; Li et al., 2017a; Zhuang et al., 2011). In 1994, it was first determined that Arabidopsis thaliana contains members of the AP2/ERF family (Jofuku et al., 1994). Based on an analysis of the entire genome, the majority of higher plants also contain members of the AP2/ERF superfamily of factors. The four ERF proteins were isolated from tobacco in 1995 and designated as ethylene-sensitive binding proteins: ERF1, ERF2, ERF3, and ERF4. Further analysis revealed that they were involved in the process of ethylene-induced disease (Shinshi, Usami & Ohme-Takagi, 1995). ABI4 and OsAP2-39 bind to target genes to control seed spouting and plant growth. Seed development may be impacted through the control of glucose metabolism by AP2/ERF transcription factors (Shu, Zhou & Yang, 2018). Through interactions with AFP1 in the control of ABI4 expression, the proteins of the Arabidopsis OXS3 family suppress ABA signaling (Xiao et al., 2021). The nucleus-localized AP2/ERF transcription factor PatDREB is a transcriptional activator that binds to PatPTSpro and regulates patchoulol biosynthesis by modulating gene expression (Chen et al., 2022). This factor affects patchoulol production and controls gene expression. In addition, the AP2/ERF gene family is implicated in plant secondary metabolism. For example, RhERF113 in Rose can increase CTK levels in floral tissues, which can delay the ethylene-induced senescence of the flowers (Khaskheli et al., 2018). A previous study used the yeast single hybrid method to screen ORCA2 and ORCA3 from the vinblastine flower and found that they can increase the expression of the isordixin synthase gene, thereby promoting the vinblastine precursor and increasing the accumulation of vinblastine (Menke et al., 1999). The overexpression of AaERF1 and AaORA enhances CYP71AV1, which improves and enhances the artemisinin biosynthetic pathway in Artemisia annua (Lu et al., 2013). EREB58 can improve the disease resistance of maize by affecting the synthesis of sesquiterpenes (Li et al., 2015). CitAP2.10 can affect the synthesis of (+)-valencene in ‘Newhall’ orange by activating CsTPS1 (Shen et al., 2016). It is reported that LcERF19 is involved in the up-regulation of geranial and neral biosynthesis by activating the expression of LcTPS42, providing an approach to improve the flavor of tomatoes and other fruits (Wang et al., 2022).

Dendrobium officinale Kimura et Migo is an important medicinal plant containing polysaccharides, alkaloids, terpenoids, and a variety of active ingredients with medicinal value (Chen et al., 2021). The associated TPS (Terpene Synthases) gene functions have also been sequentially confirmed with the finding of the terpenoid synthase gene family in D. officinale. DoTPS10, for instance, has been demonstrated to play a role in the production of linalool (Yu et al., 2020). The regulation of the TPS gene by transcription factors has also been found in D. officinale. DobHLH4 is involved in the biosynthesis of linalool during D. officinale flower development by promoting the expression of DoTPS10 (Yu et al., 2021). The biosynthesis of terpenoids in D. officinale is also regulated by the MYB family of transcription factors. Previous research has shown that DoMYB26 and DoMYB31 can positively regulate the expression of DoECS, while DoMYB29 cannot regulate the expression of DoECS, thus affecting the synthesis of terpenoids (Lv et al., 2022). Previous studies have also reported multiple promising bioactive effects of D. officinale as an anti-aging, anti-tumor, cardioprotective, gastrointestinal protective, and anti-diabetes immunomodulatory (Chen et al., 2021). AP2/ERF transcription factors are involved in many aspects of plant development and metabolism, but the roles of AP2/ERF transcription factors in D. officinale are currently unknown.

This study used D. officinale genomic and transcriptome data analysis, as well as categorization, phylogeny, and expression pattern analysis to identify DoAP2/ERFs. DoPAES-related transcription factors were identified using a correlation analysis. This study provides new insights on the function of the AP2/ERF transcription factors in terpene biosynthesis for D. officinale.

Materials and Methods

Plant material

All of the D. officinale specimens and protocorm-like bodies (PLBs) used in this experiment were obtained from the lab at Anhui Agricultural University. D. officinale tissue culture seedlings were grown in an artificial climate box at 26 °C in a 12 h/12 h light cycle (Li et al., 2021). The induction methods used in this study for protocorms of Dendrobium officinale were based on those outlined by Lv et al. (2022). Tobacco was grown in a greenhouse at 25 °C in 16 h/8 h (light/dark) photoperiod conditions. Tobacco leaves that were 4–5 weeks old were used for the injection.

DNA isolation, RNA isolation, cDNA synthesis

A complete D. officinale plant was used for RNA and DNA extraction. The samples were immediately frozen in liquid nitrogen after collection and stored at −80 °C. The DNA extraction and RNA extraction kits used in the experiments were obtained from Chengdu Bafet Biotechnology Co, Chengdu, China.

Chemical profiling

The GC-MS method described by Li et al. (2021) was used to analyze the treated D. officinale PLBs. The volatile compounds of the PLBs were collected by headspace SPME and adsorbed by 75 µm CAR/PDMS fiber (Sigma-Aldrich, St. Louis, MO, USA) at 25 °C for 1 h. All captured volatile compounds were subsequently thermally desorbed and transferred to an Agilent 5975-6890N gas chromatography-mass spectrometry (GC-MS) instrument (Agilent Technologies, Santa Clara, CA, USA). The detected volatile compounds were identified and qualitatively analyzed using the National Institute of Standards and Technology (NIST) 2011 standard library (Xu et al., 2019a; Yahyaa et al., 2015).

Heterologous expression and enzyme activity assays

The pMAL-c2x vector containing the DoPAES gene was transformed into Escherichia coli BL21, which was cultured on the LB agar plate containing 100 mg/L ampicillin to achieve the hetero expression of DoPAES. The bacterial solution was shaken at 37 °C until OD600 = 0.6, then the expression of bacterial cell fusion protein was induced by adding 0.5 mM isopropyl-β-D-thiogalactoside (IPTG) at 16 °C. The bacterial solution was induced overnight at 16 °C and centrifuged at 12,000 r/min and 4 °C for 10 min. Part of the bacterial body weight was suspended in 1×PBS buffer, added into 5×SDS-PAGE loading buffer, mixed, and placed at 100 °C for 10 min to obtain an electrophoretic sample. The samples were then subjected to sodium dodecyl sulfate-polyacrylamide gel (SDS-PAGE) electrophoresis. The remaining bacteria were resuspended in 3 mL of pre-cooled buffer (20 mmol/L Tris pH 8.0, 10 mmol/L DTT, 5 mmol/L Na2S2O5, and 10% glycerol) to obtain the crude enzyme solution and then sonicated (sonication power ratio 15%, crushing 8 s, interval 45 s, crushing 10 min). The crushed protein solution was centrifuged at 4 °C for 10 min at 12,000 r/min and the supernatant was collected to obtain the crude protein.

The recombinant protein was assayed in vitro by headspace extraction combined with GC-MS and 500 μL of target gene crude protein supernatant, 50 mmol/L Tris pH 7.5, 10 mmol/L MgCl2, 10 μmol/L MnCl2, and 1 μL FPP were added for a total volume of 1 mL. After incubating at 30 °C for 30 min, the sample was kept at room temperature for 1 h and then GC-MS detection was performed on the sample. The adsorbed compounds were analyzed using GC-MS according to the methods outlined by Li et al. (2021).

Identification of DoAP2/ERF genes

The AP2/ERF gene family sequences in A. thaliana were obtained from the Arabidopsis Information Resource (TAIR) website (https://www.arabidopsis.org/), and used to identify the AP2/ERF gene family members in D. officinale. Using the AP2/ERF sequences of A. thaliana as query sequences, a BLAST analysis was performed with the D. officinale genome using the Cluster Database at High Identity with Tolerance (CD-HIT; http://www.bioinformatics.org/cd-hit/) to remove redundant sequences when the thresholds were greater than 95%. The candidate sequences were further validated using PFAM (https://pfam.xfam.org/) to identify the AP2/ERF transcription factor family members in D. officinale. The physicochemical properties of each DoAP2/ERF protein were determined using the ProtParam online tool (https://web.expasy.org/protparam).

Construction of phylogenetic trees

IQTREE was used for phylogenetic evolutionary tree construction and the output result files were inputted into the Interactive Tree Of Life (iTOL; https://itol.embl.de/) website for tree beautification.

Gene structure, conserved motif prediction, and GO analysis

Multiple predictions of conserved motifs in DoAP2/ERFs proteins were made. Motif elicitation (https://meme-suite.org/meme/) was expected to be maximized, and the maximum number of motifs was adjusted to 10. DoAP2/ERF gene exons and introns were analyzed using the GSDS 2.0 Gene Structure Display Server 2.0 (http://gsds.gao-lab.org/) online tool to determine their location and quantity. DoAP2/ERF proteins were prepared using BLAST2GO’s non-redundant protein database software, and the DoAP2/ERF protein sequences were compared using NCBI’s protein blast tool, BLASTP. The GO annotation for each DoAP2/ERF was then retrieved from Map GO and classified using the WEGO (https://wego.genomics.cn/) online tool.

Heat map analysis

Transcriptome data of D. officinale (PRJNA348403) were used to obtain the gene expression data in eight tissues of D. officinale: column, sepal, white part of the root, green root tip, stem, leaf, lip, and buds. The transcriptional level of the DoAP2/ERF family of genes was analyzed using Tbtools Visualization. Based on the transcriptome data, a Pearson correlation coefficient calculation was performed on DoAP2/ERF family genes and DoPAES, and the calculated results were visually analyzed using the Cytoscape software.

Subcellular localization analysis

The DoPAES coding sequence was fused with the green fluorescent protein (GFP) and cloned into the pCAMBIA1305 vector. Using Arabidopsis protoplasts as an experimental material, a PEG-mediated transformation of Arabidopsis protoplasts was used to examine the subcellular localization of DoPAES. Fluorescence images of green fluorescent protein (GFP) and chlorophyll signal distribution in Arabidopsis protoplasts were observed under the IX81 Olympus confocal microscope at 488 and 640 nm. The control group was the pCAMBIA1305-GFP empty vector.

The coding sequences of DoAP2/ERFs were fused with a green fluorescent protein (GFP) and cloned into the pCAMBIA1305 vector, and the coding sequences of DoAP2/ERFs with a green fluorescent protein (GFP) were cloned into the pCAMBIA1305 vector and then transformed into Agrobacterium tumefaciens competent (GV3101). The methods used for injecting the suspension into tobacco leaves were based on methods used in previous studies (Ke et al., 2019). After one day in darkness followed by two days in normal light conditions, the injected leaves were observed under a confocal laser scanning microscope (LCSM).

Yeast one-hybrid assay

The transcriptional activity of DoAP2/ERFs was verified using the yeast-one-hybrid system, and the promoter of DoPAES was cloned into the pAbAi vector as bait (DoPAES-pAbAi). The full-length DoAP2/ERFs were fused into the pGAL4 activation vector (AD-pGADT7) to generate DoAP2/ERFs-pGADT7 as prey. They were then co-transformed into Y1H yeast cells and grown. The positive clones were screened on defective SD medium.

Dual-luciferase assay (Dual-LUC)

The DoPAES promoter was ligated into the pGreenII 0800-LUC reporter vector. These TFs were embedded into the pGreenII62-SK vector. The recombinant plasmid was transformed into Agrobacterium tumefaciens competent cells (GV3101) and then injected into the leaves of N. benthamiana and assessed for LUC activity using enzyme markers.

Transgenic expression and volatile terpenoid analysis

The RNA interference fragments of DoAP2/ERFs were connected with the pCAMBIA1305 vector and DoAP2/ERFs was also connected with the pCAMBIA1305 vector. The positive vector was transformed into competent cells of Agrobacterium tumefaciens strain GV3101 to prepare transiently transformed PLBs. PLBs were cultured in 25 °C liquid medium for three days in dark conditions and treated with Agrobacterium tumefaciens infection solution OD600 = 0.6 for 10 min. PLBs were then cultured in co-medium at 25 °C in dark conditions for 3 days. Agrobacterium tumefaciens was then removed by washing with sterile distilled water and sterile water containing 0.3 mg/L Temetine. Finally, the PLBs were exposed to a selective medium containing 0.1 g/L hygromycin and 0.3 mg/L Temetine for light exposure for 3 days (Gnasekaran et al., 2014; Ma et al., 2020). GC-MS was used to detect the changes of terpenoids (Li et al., 2021). Ethyl-decanoate (200 μg/kg FW·h) was added to the samples as the internal standard. A quantitative analysis was performed according to the peak area of the internal standard, and three repeated analyses were performed for each sample. The contents of the volatiles were expressed as μg/kg FW·h.

Results

GC-MS analysis of volatile terpenes from D. officinale PLBs

Volatile terpenes from D. officinale PLBs were detected by HS-SPME-GC-MS. Only eight sesquiterpenes were detected in the protocorms: β-Elemene (10%), β-Patchoulene (11%), Caryophyllene (31%), γ-Elemene (7%), α-Guaiene (4%), Humulene (6%), Longifolene (4%), and γ-Patchoulene (27%; Fig. 1). Patchoulene content was the highest in D. officinale PLBs, followed by Caryophyllene.

Figure 1 GC-MS analysis of volatile terpenes from D. officinale PLBs.

Percentage of each component: β-Elemene, 10%; β-Patchoulene, 11%; Caryophyllene, 31%; γ-Elemene, 7%; α-Guaiene, 4%; Humulene, 6%; Longifolene, 4%; γ-Patchoulene, 27%.

Identification and analysis of DoPAES in D. officinale

Based on the transcriptome data, the whole sequence of DoPAES was annotated. DoPAES had an overall length of 1,653 bp and could encode 550 amino acids. The DoPAES protein had a relative molecular weight of 64.9 kDa and an isoelectric point of 5.42. A phylogenetic tree analysis revealed that DoPAES is a member of the TPS-a subfamily and clusters with the plant TPS gene in charge of sesquiterpene synthesis (Fig. 2). Both DoPAES and tree members share the conserved domains of terpenoid synthases, including RXR, DDXXD, and NST/DTE (Fig. 3), which was assumed to be involved in the synthesis of β-patchoulene and was named DoPAES (GenBank: MT512059.1). The most closely-related gene, with 68.6% similarity, was the α-humulene synthase in Phalaenopsis equestris.

Figure 2 Phylogenetic tree analysis of DoPAES and genes in the TPS gene family.

The phylogenetic tree was drawn using the IQTREE program from an alignment of full-length DoPAES with other plant TPSs. The TPS family was subdivided into six subfamilies, designated TPS a-TPS-g.

Figure 3 Multiple sequence alignments of the deduced amino acid sequences among DoPAES and other plants.

Alignment of deduced amino acid sequences of DoPAES with Phalaenopsis equestris alpha-humulene synthase-like (XP_020584121.1), Vanda hybrid cultivar sesquiterpene synthase (ABX57720.1), Apostasia shenzhenica (-)-germacrene D synthase (PKA52858.1), Oncidium hybrid terpene synthase (QAA95893.1), and Phoenix dactylifera alpha-humulene synthase-like (XP_026663090.1). The conserved RxR, DDxxD, and NST/DTE motifs are indicated by red boxes. Numbers indicate the position of the last amino acid in each line of proteins.

The tertiary spatial structure of the DoPAES protein was analyzed and predicted. The homology modeling of DoPAES was performed using 5-achiral aristolodene synthetase (4RNQ1.A) from tobacco. The homological similarities between DoPAES and the template protein sequence was 36.82%. The protein template was sesquiterpene synthetase, and the conformation of DoPAES was similar to the terpene synthetase of other species (Fig. S1).

To determine the subcellular localization of DoPAES, a vacant plasmid was fused with a GFP tag and transiently expressed in Arabidopsis protoplasts, and the results showed that the GFP-fused DoPAES exhibited cytoplasmic localization (Fig. 4).

Figure 4 Subcellular localization of DoPAES in Arabidopsis thaliana protoplast.

GFP was used as control. Scale bars indicate 10 µm.

Expression and enzyme activity analysis of the DoPAES protein

Volatile sesquiterpenes were mainly synthesized through the mevalonate (MVA) pathway. The successfully-constructed recombinant plasmid, pET32a-DoPAES, was introduced into BL21 expressing strains. The products were then detected by GC-MS after adding substrate FPP. The results showed, and in vitro enzyme activity confirmed, the catalytic activity of DoPAES and its role in sesquiterpene biosynthesis. The GC-MS analysis showed that the enzyme reaction products contained terpenoid β-patchoulene (Fig. 5). These results indicate that DoPAES can specifically catalyze the synthesis of β-patchoulene.

Figure 5 HS-SPME-GC-MS analysis of products produced by recombinant DoPAES protein in vitro.

GC-MS peaks of volatile compounds were detected using extracts of pMAL-DoPAES cells with the addition of FPP. (A) The GC-MS peaks of the pMAL-DoPAES product after addition of FPP. (B) and (C) The mass of β-patchoulene detection and corresponding spectral library.

The volatile terpene components of D. officinale were previously detected and compared with the active products of DoPAES. Based on the retention time (RT) and ion peak of the target, β-patchoulene was found to be the main enzymatic product of DoPAES in D. officinale PLBs.

Identification of the D. officinale AP2/ERF gene family

The A. thaliana AP2/ERF amino acid sequence was used as a query to blast against the D. officinale genome to identify the members of the AP2/ERF family. As a result, 122 AP2/ERF candidate genes were screened out and 111 non-redundant sequences were identified and named DoAP2/ERF1-111. The 111 DoAP2/ERF genes were initially divided into three subfamilies based on the classification of structural domains (Fig. 6). The RAV subfamily (which contains one AP2 domain and one B3 domain), the ERF subfamily (which contains one AP2 domain), and the AP2 subfamily (which contains two AP2 domains). A phylogenetic analysis showed that the 111 DoAP2/ERFs could be divided into these subfamilies as follows: 14 in AP2, three in RAV, 92 in ERF, and two in the soloist subfamily, according to the priority classification rules of A. thaliana AP2/ERF TFs.

Figure 6 Phylogenetic analysis of AP2/ERF proteins in Dendrobium officinale.

A total of 111 AP2/ERF proteins from D. officinale were selected to construct the phylogenetic tree. Cyan represents the ERF subgroup, blue represents the AP2 subgroup, purple represents the RAV subgroup, and dark blue represents the soloist subgroup.

The physicochemical characteristics of the DoAP2/ERF proteins were also examined. The results showed DoAP2/ERF10 was responsible for encoding the largest number of amino acid termini (661), whereas DoAP2/ERF17 displayed the highest molecular weight value, reaching 71.34. DoAP2/ERF17 was also identified as the heaviest of the 111 transcription factors because of its overall atom count of 9,736. The theory pI of the DoAP2/ERFs ranged from 4.35 (DoAP2/ERF98) to 10.1 (DoAP2/ERF96). All of the DoAP2/ERF proteins had negative values for the grand average of hydropathicity (GRAVY), indicating that they are all hydrophilic proteins. All of the instability indices were high, indicating that these proteins are unstable (Table S1). These findings suggest that DoAP2/ERF proteins might have distinct functions in various cell sites.

Phylogenetic analysis of the AP2/ERF gene family of D. officinale

A phylogenetic tree was built using 111 DoAP2/ERF and 147 A. thaliana AP2/ERF transcription factor families. Based on the results, the ERF subgroup, with the greatest number of family members, was further divided into 10 different branches, denoted I through X. The AP2 subgroup consisted of 14 D. officinale and 18 A. thaliana members; the RAV subgroup contained three D. officinale and six RAV subgroup members of A. thaliana. The results also showed that the majority of the superfamilies and subfamilies in the evolutionary tree were produced by A. thaliana and D. officinale AP2/ERF family member genes, showing that AP2/ERF family member genes may be homologous and may have evolved from the same ancestor.

Conserved motif analysis of the AP2/ERF gene family of D. officinale

The MEME online tool was used to characterize the potentially conserved motifs of the 111 DoAP2/ERF sequences from the amino acid sequences of DoAP2/ERF members (Fig. 7). Ten conserved motifs were identified, numbered 1 through 10 (Fig. S3). The majority of each group’s members had similar motifs. Although each subfamily of transcription factors had a distinct length, they all had conserved motifs that were largely the same. For instance, motif 1 was found in all 111 DoAP2/ERF family genes, proving that it was a crucially conserved pattern in AP2/ERF genes. The protein sequences of transcription factors contained several conserved motifs, which may function as potential DNA-binding sites to help regulate gene expression.

Figure 7 Phylogenetic relationships of conserved protein motifs in AP2/ERF proteins in D. officinale.

On the left is the phylogenetic tree constructed with 111 AP2/ERF proteins from D. officinale. On the right is the architecture of conserved protein motifs in different subfamilies. The colored boxes indicate the different motifs, as listed on the right of the figure.

Gene structure analysis of the D. officinale AP2/ERF gene family

The locations of exons and introns of the 111 DoAP2/ERF sequences were analyzed using the GSDS online tool to further explore the function of the D. officinale AP2/ERF gene family (Fig. 8). The phylogenetic topological categorization of gene families was also supported by gene structure research. The number of introns in AP2/ERF family genes varied from one to nine in earlier reports on A. thaliana and Syntrichia caninervis (Jofuku et al., 1994; Li et al., 2017a). The exon and intron architectures of DoAP2/ERF genes were similar within the same subclass. The results showed that the majority of AP2 subfamily members had numerous exon and intron distributions. Although there were differences in the number, the positions of the introns and the exon regions were relatively conserved. The intronic sections were large and the exonic regions were shorter. Similar results were observed in ERF subfamily members, with the exception that each member of groups X and VII had two exonic regions, one at either end and one in the middle, respectively. Most ERF subfamily members only had one exonic area and no intronic regions outside of this region. The AP2 subfamily’s exon distribution was comparable for the soloist and had several introns. All AP2/ERF family members had the same conserved region at their N-terminus, which may be crucial to their functions.

Figure 8 Phylogenetic relationships and intron pattern of AP2/ERF proteins in D. officinale.

On the left is the phylogenetic tree constructed with 111 AP2/ERF proteins from D. officinale. On the right is the predicted exon-intron structures. The yellow boxes and black lines represent exons and introns, respectively.

GO analysis of the AP2/ERF gene family of D. officinale

The 111 DoAP2/ERF proteins from D. officinale underwent GO enrichment analysis. A total of 24 GO elements were enhanced when the P-value was less than 0.05. Blast2GO software was used to annotate the 111 D. officinale AP2/ERF proteins in 24 GO categories, visualize the GO terms and predict their participation in various biological processes, and WEGO was used to classify the GO terms (Fig. S4). Based on similarities in amino acid sequences, DoAP2ERF protein sequences were divided into three main groups: cellular component (CC), molecular function (MF) and biological process (BP), with 18 proteins belonging to the “biological process” category, four proteins grouped into “cellular component,” and two proteins classified as “molecular function.” The three most enriched items in the functional category of cellular components were cells, cell parts, and organelles. Transcription regulator activity in the biological process category of molecular function had the largest enrichment degree. Metabolic activities, cellular processes, biological control, and regulation of biological processes were the four entries with the highest enrichment. The GO analysis showed that the DoAP2/ERF transcription factor family was involved in most of the life processes of plant cells.

Heat map analysis of the AP2/ERF gene family of D. officinale

Transcriptome sequencing (RNA-Seq) data from eight different tissues of D. officinale were downloaded from the National Center for Biotechnology Information (NCBI) using the BioProject accession number PRJNA348403. The expression patterns of DoAP2/ERF genes are shown in Fig. S5. The majority of the family genes, including DoAP2/ERF101, DoAP2/ERF10, and DoAP2/ERF65, had very low expression levels in these eight tissues, with some having no expression or expressing in just select tissues. A small number of genes, including DoAP2/ERF07, DoAP2/ERF26, DoAP2/ERF27, DoAP2/ERF37, and DoAP2/ERF29, were substantially expressed in different tissues. Overall, the majority of the AP2/ERF family of genes were expressed in the flower tissue.

The expression patterns of DoAP2/ERF and DoPAES

In plants, the AP2/ERF transcription factor family is crucial. A gene co-expression analysis was carried out based on transcriptome data to investigate the spatiotemporal regulation of DoAP2/ERF and terpenoid synthases DoPAES. The highly-correlated DoAP2/ERF89 and DoAP2/ERF47 were chosen based on the findings of the correlation screening (Fig. 9), as they may help control the expression of DoPAES.

Figure 9 Co-expression network diagram of terpenoid synthesis-related genes and DoAP2/ERF transcription factors.

Screening DoAP2/ERF89 and DoAP2/ERF47 show a high correlation with DoPAES.

Identification and characterization of DoAP2/ERFs

DoAP2/ERF89’s full-length cDNA was 801 bp long, encoding 266 amino acid sequences, with a molecular weight of 28.62 kDa. DoAP2/ERF47’s full-length cDNA was 981 bp long, encoding 326 amino acid sequences, with a molecular weight of 35.89 kDa. In accordance with the prior evolutionary tree, the protein sequence analysis revealed that DoAP2/ERF47 contained AP2 and B3 binding domains and belonged to the RAV subfamily, whereas DoAP2/ERF89 contained an AP2-binding domain and belonged to the ERF subfamily.

Subcellular localization analysis

DoAP2/ERF89 and DoAP2/ERF47 were cloned into the pCAMBIA-1305 vector and transformed into tobacco plants using Agrobacterium tumefaciens strain GV3101. The results showed that DoAP2/ERF89-GFP and DoAP2/ERF47-GFP were only found in the nucleus and displayed blue after fusion in the same cell, whereas the GFP fluorescence of the empty vector was distributed throughout tobacco leaf cells (Fig. 10). These results demonstrated the localization of the DoAP2/ERF89 and DoAP2/ERF47 proteins in the nucleus.

Figure 10 Subcellular localization of DoAP/ERFs.

Nuclear localization of DoAP/ERF89 and DoAP/ERF47 in tobacco leaves. The scale bar indicates 20 μm.

Analysis of the interaction between DoAP2/ERF and DoPAES

In order to determine whether DoAP2/ERF89 and DoAP2/ERF47 were involved in the regulation process of DoPAES, the promoter of cloned DoPAES was linked to the pAbAi vector and verified by yeast one-hybrid assay (Y1H). The results showed that the bait strain co-expressing DoAP2/ERF89, DoAP2/ERF47, and proDoPAES grew well in SD/-Leu medium containing the antibiotic Aureobasidin A (Fig. 11), indicating that DoAP2/ERF89 and DoAP2/ERF47 could bind to the promoter of DoPAES.

Figure 11 Binding of DoAP2/ERFs to DoPAES promoters.

Yeast one-hybrid assays reveal that DoAP2/ERF89 and DoAP2/ERF47 can bind to the promoters of DoPAES. The yeast cells were grown on an SD/-Ura/-Leu+200 ng/mL AbA.

The previous results showed that the expression trend of DoAP2/ERF47 and DoAP2/ERF89 were consistent with DoPAES in D. officinale, suggesting that these TFs may regulate the expression of DoPAES. Dual-LUC analysis further confirmed that these TFs regulated the expression of DoPAES. The DoPAES promoter was placed into the pGreenII0800-LUC vector, while these transcription factors were embedded into the pGreenII62-SK vector. The results revealed that DoAP2/ERF89 considerably activated the DoPAES promoter compared to the control group, although DoAP2/ERF47 activation was less pronounced (Fig. 12).

Figure 12 Binding and interaction activity of DoAP2/ERF proteins.

Transcription interaction activity between DoAP2/ERF89, DoAP2/ERF47, and DoPAES were assessed in tobacco leaves. Empty vector of pGreen II 62-SK was used as a control.

DoAP2/ERF89 promotes the formation of terpenoids from D. officinale

Volatile terpene concentrations in the control group (CK), DoAP2/ERFs-OE, and DoAP2/ERFs-RNAi were determined through the transient expression of candidate genes in D. officinale PLBs. The results showed that overexpression of DoAP2/ERF89 increased the content of β-patchoulene compared with CK. The amount of β-patchoulene in PLBs was decreased when DoAP2/ERF89 was silenced (Fig. 13). DoAP2/ERF47 also had an impact on terpenoid synthesis, but not on the synthesis of β-patchoulene. These findings indicate that the synthesis of β-patchoulene in D. officinale is regulated by DoAP2/ERF89.

Figure 13 GC-MS analysis of transient transformation of D. officinale PLBs.

CK, Control group; OE, DoAP2/ERF89 overexpression; RNAi, DoPAE RNA interference instantaneous transformation of protocorm of D. officinale was analyzed by GCMS. IS, Internal Standard; 1, β-Elemene; 2, β-Patchoulene; 3, Caryophyllene; 4, γ-Elemene; 5, α-Guaiene; 6, Humulene; 7, Longifolene; 8, γ-Patchoulene.

Discussion

Terpenoids, one of the abundant compounds released by flowered plants, play a variety of roles in the biological processes of plants (Muhlemann, Klempien & Dudareva, 2014). For example, the common monoterpene β-ocimene released by plants and flowers plays a key role in attracting pollinators (Farré-Armengol et al., 2017). Linalool in strawberry aids in the plant’s resistance to external damage (Xu et al., 2019b). β-Patchoulene is prevalent in plants and plays an important role in anti-inflammation, anti-infection, and protection against insect pests (Zhang et al., 2016; Pu et al., 2019). D. officinale is an important medicinal plant because of its terpenoids. The terpene metabolism profile of D. officinale PLBs was constructed in this study and the results showed that β-patchoulene had the highest content and accounted for the largest proportion. The genomic and transcriptomic analysis of DoPAES revealed the highest homology of DoPAES with sesquiterpene synthase. The functional analysis verified that DoPAES was involved in the synthesis of β-patchoulene in D. officinale.

Secondary metabolic pathways in plants are intricate and influenced by a variety of factors. Terpenoid genes are regulated at several stages, including transcription and post-translation. The family of transcription factors known as AP2/ERF is broadly distributed in plants. Several AP2/ERF transcription factors have previously been shown to play a role in the control of terpenoid synthesis. For instance, in sweet orange, CitERF71 can directly bind to the promoter of the CitTPS16 terpene synthase gene, participating in the regulation of e-geraniol production (Li et al., 2017b). SmERF128 can positively regulate the biosynthesis of diterpene tanshinone by activating the expression of SmCPS1, SmKSL1, and SmCYP76AH1 in Salvia miltiorrhiza (Zhang et al., 2019).

This study identified the AP2/ERFs in D. officinale, and a correlational analysis of the transcriptome data revealed that the expression patterns of DoAP2/ERF89 belonged to subfamily IX and DoAP2/ERF47 was classified as part of the RAV subfamily. Previous research has demonstrated that IX subsets play a vital role in plant-specific metabolic pathways (Paul et al., 2020). These findings imply that these transcription factors may play essential roles in the terpenoid biosynthesis of D. officinale.

Further experiments demonstrated that DoAP2/ERF89 and DoAP2/ERF47 can bind to the promoter of DoPAES and DoAP2/ERF89 can activate the synthesis of β-patchoulene by regulating DoPAES. These results show that the transcription factors from the AP2/ERF family play important roles in the control of secondary metabolic pathways and the accumulation of terpenoids in plants. Similarly, SmERF1L1 is involved in the regulation of salvianol biosynthesis in Salvia miltiorrhiza (Huang et al., 2019). AaERF1 and AaERF2 are highly expressed in inflorescences and can positively regulate artemisinin biosynthesis in A. annua (Yu et al., 2012). In this study, yeast one-hybrid and dual-luciferase assays demonstrated that DoAP2/ERF89 and DoAP2/ERF47 can bind to the promoter of DoPAES. However, the transient expression of PLBs showed that only DoAP2/ERF89 can positively regulate the synthesis of β-patchoulene in D. officinale.

Conclusion

D. officinale is an important medicinal plant in China, and its terpenoids are one of its main medicinal components. Previous research has shown that AP2/ERF genes are closely related to secondary metabolism, growth, and development in plants. This study constructed the terpene metabolic profile of D. officinale PLBs and found a high content of patchoulene. The DoPAES gene was shown to control the biosynthesis of β-patchoulene in D. officinale. A total of 111 members of the DoAP2/ERF family were identified. A correlation analysis revealed that DoAP2/ERF89, DoAP2/ERF47, and DoPAES had a relatively high correlation index. Yeast one-hybrid and Dual-luciferase experiments verified that DoAP2/ERF89 and DoAP2/ERF47 can regulate the expression of DoPAES by interacting with its promoter, but only DoAP2/ERF89 is able to positively regulate DoPAES in β-patchoulene synthesis. These findings provide new insights into the function and transcriptional regulation of volatile terpene synthase genes in medicinal plants.

Supplemental Information

Supplemental Information 1 Homologous modeling of DoPAES proteins.

The protein template is sesquiterpene synthetase, and the conformation of DoPAES is similar to that of terpene synthetase of other species.

Click here for additional data file.

Supplemental Information 2 Phylogenetic analysis of AP2/ERF proteins in D. officinale and A. thaliana.

In total, 111 AP2/ERF proteins from D. officinale, 147 AP2/ERF proteins from A. thaliana were selected to construct the tree. Ⅰ–Ⅹ indicated the divided subfamilies according to the categories of AP2/ERF proteins in A. thaliana.

Click here for additional data file.

Supplemental Information 3 Sequence logos of the AP2/ERF repeats.

Motif 1–10 represents identity based on multiple alignment analysis of 111 DoAP2/ERF proteins.

Click here for additional data file.

Supplemental Information 4 GO analysis of the AP2/ERF gene family in D. officinale.

The 111 DoAP2/ERFs protein sequences are divided into three categories based on amino acid similarity, namely cellular components (CC), molecular function (MF), and biological processes (BP).

Click here for additional data file.

Supplemental Information 5 Expression patterns of 111 DoAP2/ERFs from D. officinale in different tissues.

The heatmap was generated using TBtools and the FPKM values of D. officinale genes were evaluated and normalized based RNA-seq data from NCBI SRA database. Differential expression pattern of 111 annotated DoAP2/ERFs in various tissues, including roots, root tips, stems, leaf, lip, buds, and sepal.

Click here for additional data file.

Supplemental Information 6 Physical and chemical properties of AP2ERF protein from Dendrobium officinale.

Click here for additional data file.

Supplemental Information 7 Raw data.

GC-MS analysis of transient transformation of D. officinale PLBs.

Click here for additional data file.

Supplemental Information 8 Heatmap raw data of DoAP2/ERFs.

Click here for additional data file.

Additional Information and Declarations

Competing Interests

Author Contributions

Data Availability

The authors declare that they have no competing interests.

Decong Li conceived and designed the experiments, performed the experiments, prepared figures and/or tables, and approved the final draft.

Lin Liu analyzed the data, prepared figures and/or tables, and approved the final draft.

Xiaohong Li performed the experiments, prepared figures and/or tables, and approved the final draft.

Guo Wei conceived and designed the experiments, performed the experiments, prepared figures and/or tables, authored or reviewed drafts of the article, and approved the final draft.

Yongping Cai conceived and designed the experiments, authored or reviewed drafts of the article, and approved the final draft.

Xu Sun analyzed the data, prepared figures and/or tables, and approved the final draft.

Honghong Fan conceived and designed the experiments, authored or reviewed drafts of the article, and approved the final draft.

The following information was supplied regarding data availability:

The raw measurements are available in the Supplemental Files.

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
