# Peer review of "DoAP2/ERF89 activated the terpene synthase gene DoPAES in Dendrobium officinale and participated in the synthesis of β-patchoulene"

_PeerJ, doi:10.7717/peerj.16760_

## Round 0.1 · original submission · Major Revisions

Your manuscript was reviewed by three experts in the field. The reviewers found the work interesting but raised several issues which need to be addressed properly. The reviewers provide detailed comments in their reviews and point out the areas where the manuscript needs to be improved. I also read the manuscript carefully and largely agree with the reviewers’ comments. In particular, extensive rewriting of this manuscript with a clear hypothesis and language is critically important for publication in PeerJ.

·

Basic reporting

In the current manuscript, the authors set out to identify sesquiterpene synthetase-β-patchoulene synthase (DoPAES) gene and 111 AP2/ERF family members in in Dendrobium officinale. Further they analyze tissue-specific expression pattern and gene co-expression analysis of DoAP2/ERF family members and by Yeast one-hybrid assays and double luciferase experiments showed that DoAP2/ERF89 and DoAP2/ERF47 regulated the expression of DoPAES and DoAP2/ERF89 could positively regulate the biosynthesis of β-Patchoulene. Manuscript is as such well-designed experimentally and analyzed.

1.Line 69: Author should write the full form of TaPS and write in context to manuscript as it comes abruptly, author should include some background to it.

2.It would be better if Fig 1. % of identified compound indicated in Pie diagram.

3.Figure 13 should be supplemented with MS spectra as it important piece of result to claim that DoAP2/ERF89 is involved in the synthesis of β-patchoulene in D. officinale and author should comment on the β-patchoulene level in CK, OE and RNAi and their comparison. I am wondering is whether any internal standard included all three runs to compare? Authors should explain it clearly and incorporate in manuscript.

Experimental design

Material and method section needs to be improvised
1.Line 147: How authors performed the enzyme activity assay, it is not clear from the material and method section, author should rewrite the enzyme activity assay section.
2.Line 213: Authors should elaborate the volatile terpenoid analysis in material and methods section for CK, OE and RNAi and how they compare the level of β-Patchoulene?

Validity of the findings

No comments

Additional comments

1.Line 168: Use the online GSDS 2.0 Rephrase the sentence and “Location” word should start with small alphabet
2.Line 171: What is White matter sequence? Authors should rephrase the statement
3.Line 176: Authors should remove Column; “ sepel” should be written as “sepal” “Buds” should be “buds”
4.Line 202: Line 219; B. benthamiana should be N. benthamiana
5.Line 243: Authors should include the Full form of MVA Pathway.
6.Line 392: Authors should mention the full form of “CK”
7.Line 408; 453; 454: Patchouliene should be patchoulene

Reviewer 2 ·

Basic reporting

This research explored the AP2/ERF transcription factor family in an important medicinal plant – D. officinale and identified the association between DoPAES and DoAP2/ERF from both molecular biological and bioinformatical perspectives. This work is of importance to study the metabolism of terpenoids in D. officinales. However, there are several issues that need to be addressed and the quality of this manuscript needs improvement before moving forward for publication.
In general, the writing needs big improvement. There are grammar errors and some parts don’t comply with the scientific writing standards. It’s suggested to proofread by someone who is more efficient in scientific writing or find some professional editing service.
Secondly, the structure is not clear. There is no clear logic throughout the manuscript about what’s the hypothesis of this study and how each piece of the results served as evidence stacking together to support the hypothesis (just one example, “3.1 GC-MS analysis of volatile terpenes from D. officinale PLBs” - how that result support the hypothesis?)
Introduction part needs more description of AP2/ERF family and its interaction with plant hormones.

Experimental design

Line 90-91 “The AP2/ERF transcription factors involved in regulating the biosynthesis of terpenoids” This sentence is not associated either with the above or the below.
Line 131, could you explain - why only flower tissue is used RNA-extraction?
Line 130, where the authors talked about DNA RNA extraction. However, what’s DNA isolation for? I didn’t find any results related to DNA sequencing data in the result session. Also, looks like the authors did the RNA extraction and the sequencing themselves, however, in line 338, the RNA-seq data is downloaded from a public database, which is confusing. Please clarify here.
Line 154, don’t need to mention the invisible Markov model, since the tool you’re using here is BLAST.
Line 156, could you please define what’s “redundant sequence” here? What are the thresholds to remove them?
Line 349 there is no description of how gene co-expression analysis is done.
Figure 2 the neighbor-joining method is only for clustering, which can’t be used for phylogenetic relationship analysis. You could use RAxML or IQtree.
Figure 2 tps-e/f does not belong to one monophyletic group, so why group them together?

Validity of the findings

Line 346, I think it’s worthy to list the tissue specifically expressed AP2/ERF genes and what’s are those tissues. Those are valuable information to know the tissue-specific expression of AP2/ERF family.
Figure 7 The resolution is too low to view the figure. Please provide a high-resolution figure.

·

Basic reporting

Detailed and rigorous experiments were conducted to determine the role of DoPAES in the synthesis of β-patchoulene.

I have minor comments,

1. Gene names should have been mentioned fully in the manuscript. For example, TaPS. There were several names, and other abbreviations need to be mentioned fully at the beginning.

2. Several grammatical errors (for example, repeated words in the conclusion section). The paper may benefit from close English language editing.

Experimental design

The paper is well-written, and the experimental designs are very rigorous.

Validity of the findings

Findings are discussed in detail and benefit a wide range of audiences.

---

## Round 0.2 · Major Revisions

Although authors fail to revise the manuscript properly, I would like to offer them a chance to address the concerns or comments of reviewer 2. For example, reviewer mentioned in the last review report that the neighbor-joining method is only for clustering, can't be used for phylogenetic relationship analysis. The authors responded "we used the neighbor-joining method only for clustering", however in line 182-183 "At the same time, a separate phylogenetic tree was constructed for DoAP2/ERF using the same method." - This is contradictory. Besides, there are also lots of obvious typos in the revised manuscript.

·

Basic reporting

'no comment'

Experimental design

'no comment'

Validity of the findings

'no comment'

Additional comments

Figure 2. "Phylogenetic analy" word should be deleted
Figure 5. Figure 5A should be mentioned in the figure legend and change the word “spectrogram” to spectrum

Reviewer 2 ·

Basic reporting

NA

Experimental design

NA

Validity of the findings

NA

Additional comments

NA

---

## Round 0.3 · Minor Revisions

Although the authors addressed the key comments, they did not pay attention to checking the English and typos.  The present version has numerous typographical errors throughout the manuscript. A few instances are provided below; Abstract: "transciptama", Line 104; Remove the comma after Migo, Line 139, first sentence: "We selected"? Line 144, change the reference to "The GC-MS method described by Li et al. (2021)". Therefore, I strongly encourage authors to pay close line-by-line attention to English writing, punctuation, typos, grammar issues, and sentence formation. The manuscript can not be accepted for publication in PeerJ without this improvement.

**Language Note:** The Academic Editor has identified that the English language must be improved. PeerJ can provide language editing services - please contact us at copyediting@peerj.com for pricing (be sure to provide your manuscript number and title). Alternatively, you should make your own arrangements to improve the language quality and provide details in your response letter. – PeerJ Staff

---

## Round 0.4 · Major Revisions

The reviewing manuscript does not have the changes, but editorial board looked at the tracked changes and yes authors made a number of changes, but there are far too many edits remaining for this to be handled by PeerJ. Unless authors can have the manuscript edited for English language, editorial board recommends reject. However, I would like to give a chance to the authors for the extensive language improvement by a language expert. All the changes should be done in track change mode before submission to PeerJ.

**Language Note:** The Academic Editor has identified that the English language must be improved. PeerJ can provide language editing services - please contact us at copyediting@peerj.com for pricing (be sure to provide your manuscript number and title). Alternatively, you should make your own arrangements to improve the language quality and provide details in your response letter. – PeerJ Staff

Reviewer 2 ·

Basic reporting

The authors address most of the comments from the previous reviews. However, on line 581, there is still a lack of enough details about how gene co-expression analysis is done. for example, which statistical method? Pearson correlation etc.?

Experimental design

NA

Validity of the findings

NA

---

## Round 0.5 · accepted · Accept

In my opinion, the authors have addressed the comments raised during the previous review, and the English language of the manuscript has been sufficiently improved by the authors.